# Liposome-Mediated Gene Transfer in Differentiated HepaRG™ Cells: Expression of Liver Specific Functions and Application to the Cytochrome P450 2D6 Expression

**DOI:** 10.3390/cells11233904

**Published:** 2022-12-02

**Authors:** Manuel Vlach, Hugo Coppens-Exandier, Agnès Jamin, Mathieu Berchel, Julien Scaviner, Christophe Chesné, Tristan Montier, Paul-Alain Jaffrès, Anne Corlu, Pascal Loyer

**Affiliations:** 1Institut NUMECAN (Nutrition Metabolisms and Cancer), F-35000 Rennes, France; 2Institut AGRO Rennes-Angers, F-35042 Rennes, France; 3Biopredic International, F-35760 Saint Grégoire, France; 4Univ. Brest, CNRS, CEMCA, UMR 6521, F-29238 Brest, France; 5Plateforme BiogenOuest SynNanoVect, F-44035 Nantes, France; 6Univ. Brest, INSERM, EFS, UMR 1078, GGB-GTCA, F-29200 Brest, France

**Keywords:** HepaRG™ cells, retrodifferentiation, cell cycle, transfection, lipofection, lipophosphoramidate, CYP450 2D6

## Abstract

The goal of this study was to establish a procedure for gene delivery mediated by cationic liposomes in quiescent differentiated HepaRG™ human hepatoma cells. We first identified several cationic lipids promoting efficient gene transfer with low toxicity in actively dividing HepG2, HuH7, BC2 and progenitor HepaRG™ human hepatoma cells. The lipophosphoramidate Syn1-based nanovector, which allowed the highest transfection efficiencies of progenitor HepaRG™ cells, was next used to transfect differentiated HepaRG™ cells. Lipofection of these cells using Syn1-based liposome was poorly efficient most likely because the differentiated HepaRG™ cells are highly quiescent. Thus, we engineered the differentiated HepaRG™ Mitogenic medium supplement (ADD1001) that triggered robust proliferation of differentiated cells. Importantly, we characterized the phenotypical changes occurring during proliferation of differentiated HepaRG™ cells and demonstrated that mitogenic stimulation induced a partial and transient decrease in the expression levels of some liver specific functions followed by a fast recovery of the full differentiation status upon removal of the mitogens. Taking advantage of the proliferation of HepaRG™ cells, we defined lipofection conditions using Syn1-based liposomes allowing transient expression of the cytochrome P450 2D6, a phase I enzyme poorly expressed in HepaRG cells, which opens new means for drug metabolism studies in HepaRG™ cells.

## 1. Introduction

Primary cultures of human hepatocytes, including pure cultures [1], cocultures [2] and liver organoids [3], are considered as the most pertinent in vitro hepatic cell models to study liver physiopathology and to conduct xenobiotic metabolism and toxicity. The relative shortage in liver biopsies, the constraining ethical regulation and the short lifespan of primary cultures contribute in reducing the number of laboratories that have access to these hepatocyte culture models and their use on a large scale. In addition, reproducibility of experiments may be impaired by variations in the expression of liver specific functions in hepatocytes isolated from donors varying in age, liver metabolism and treatments [1].

In contrast, human hepatoma cell lines such as HepG2 and HuH7 cells can be expanded at will but these cells express low levels of liver specific functions, especially phase I and II enzymes, which considerably reduces their biological relevance to assess metabolism and toxicity of xenobiotics [4]. For this reason, recombinant hepatoma cell lines expressing one or several drug metabolism enzymes (DME) have been established by gene transfer to enhance specific metabolic pathways in these cells [5,6,7,8,9,10]. The human hepatoma HBG [11] and HepaRG™ cell lines [12] provide alternative in vitro models to human hepatocytes. While these cells express low levels of liver specific functions during proliferation at low density, they exhibit the remarkable ability to undergo quiescence at confluency, to remain stable for several weeks at very high density and to progressively express some liver genes in various extents. The HBG cell lines derived from a hepatocellular carcinoma (HCC) include several cloned subpopulations composed exclusively of hepatocyte-like cells with homogenous morphologies [11]. In contrast, undifferentiated HepaRG™ cells are bipotent hepatic progenitor cells able to differentiate into both cholangiocyte- and hepatocyte-like cells in appropriate culture conditions [13,14]. Differentiated HepaRG™ hepatocyte-like cells express most of the DME [15,16,17] and are used for drug metabolism and toxicity evaluation of xenobiotics [18,19,20,21,22,23].

However, the expression levels of some DME, such as the cytochrome P450 2E1 (CYP2E1) [24], remain lower than those found in normal human hepatocytes, while CYP2D6 is barely detectable [15]. In order to improve metabolic activities in HepaRG™ cells, we and others have performed lentiviral transduction of proliferating progenitor HepaRG cells in order to achieve stable enforced expression of CYP2E1 [24] and CYP2D6 [25]. Based on our experiences, establishment of stable recombinant HepaRG cell lines with lentiviruses, however, requires time-consuming cell selection to isolate stable clones with homogenous expression levels of the transgene. In addition, these steps of selection often induce alterations of the proliferation and differentiation ability of the recombinant HepaRG™ cells, which lead to lower expression of the liver specific functions.

In the past years, we have set up electroporation [26,27] and lipofection [26,27,28,29] procedures to improve transient transfection of hepatoma cells including HepaRG™ progenitor and differentiated hepatocyte-like cells. While gene transfer with cationic liposomes was quite efficient to express the CYP2E1 in progenitor HepaRG™ cells, in contrast, lipofection yields to very poor transfection efficiencies in quiescent differentiated HepaRG™ cells [26,27], as previously reported for most non-dividing cell types [30,31,32].

The goal of this study was to establish a procedure for liposome-mediated gene delivery in quiescent differentiated human HepaRG™ hepatoma cells. We first identified non-toxic cationic lipids promoting efficient gene transfer in HepG2, HuH7, BC2 and progenitor HepaRG™ hepatoma cells. In contrast, liposome-based transfection of differentiated HepaRG™ cells was poorly efficient because of the low proliferation index of these quiescent cells. Thus, we engineered a mitogenic medium and a culture procedure to achieve a transient proliferation of HepaRG™ differentiated cells in order to improve their transfection efficiency by lipofection. Importantly, we characterized the changes in the expression of liver specific functions occurring during and after mitogenic stimulation. Taken advantage of the proliferation of HepaRG™ differentiated cells, we defined lipofection conditions using the cationic lipophosphoramidate Syn1-based liposomes, which allowed high expression of the GFP reporter protein as well as the CYP2D6.

## 2. Materials and Methods

### 2.1. Hepatoma Cell and Primary Human Hepatocyte Cultures

HepG2 and HuH7 cells (ATCC and European Collection of Authenticated Cell Cultures) were grown in MEM and DMEM media (Thermo Fisher Scientific, Waltham, MA, USA), respectively, both supplemented with 10% FCS (Eurobio Scientific, Les Ulis, France, and Biosera, Grosseron distributor, Coueron, France), 2% L-glutamine, 100 units/mL penicillin and 100 µg/mL streptomycin (Life Technologies, Saint Aubin, France).

HBG cells (BC2 clone, [11]) were grown in William E medium (Thermo Fisher Scientific, Waltham, MA, USA) supplemented with 10% FCS, 5 mg/L insulin (Sigma, Saint Quentin Fallavier, France), 7 × 10^−7^ mol/L hydrocortisone hemisuccinate sodium salt (Serb, Paris, France), 100 units/mL penicillin and 100µg/mL streptomycin, as previously described [11].

Progenitor HepaRG™ cells were expanded for 2 weeks using Biopredic^Int^ HepaRG™ growth medium (MIL710, Biopredic International, Saint Grégoire, France) composed of Basal hepatic cell medium (MIL700, Biopredic^Int^) supplemented with HepaRG™ Growth Medium Supplement (ADD710, Biopredic^Int^). Then, confluent quiescent cells were cultured for 2 more weeks using Biopredic^Int^ HepaRG™ Differentiation Medium (MIL720) composed of the Basal hepatic cell medium (MIL700) supplemented with HepaRG™ Differentiation Medium Supplement (ADD720 Biopredic^Int^) to further enhance the hepatocyte differentiation [13,15,17].

Human primary hepatocytes obtained from the Centre de Ressources Biologiques (CRB) of Rennes were cultured in William’s E medium supplemented with 2 mM L-glutamine, 50 IU/mL penicillin, 50 µg/mL streptomycin, 5 mg/L insulin, 10^−5^ M hydrocortisone hemisuccinate, 10% FBS and 2% DMSO.

For the lipofection of proliferating cells, 10^5^ HepG2, HuH7, HBG (BC2 clone) and HepaRG™ progenitor cells were seeded the day before the transfection in 24 well plates. For the lipofection of differentiated HepaRG™ cells, progenitor cells were seeded and expanded for 2 weeks in MIL710. Then, confluent quiescent cells were cultured for 2 more weeks in MIL720 to further enhance the hepatocyte differentiation [13,15,17]. In order to induce the proliferation, differentiated HepaRG™ cells were successively cultured in the following media: (1) HepaRG™ Growth Medium (MIL710) overnight, (2) the next day, cultures were incubated for 24 h in the HepaRG™ Growth Medium (MIL710) supplemented with the Differentiated HepaRG^TM^ Mitogenic medium supplement (ADD1001, Biopredic^Int^), (3) the third day, cultures were switched back to HepaRG™ Growth Medium Supplement (MIL710) and (4) on day 4, cells were cultured in HepaRG™ Differentiation Medium (MIL720) to inhibit the proliferation and to maintain the expression of liver specific functions.

### 2.2. Plasmids, Liposome Preparation and Lipofection

The pmax-GFP (Lonza, Levallois-Perret, France), the pEGFP-C3 (Clontech, Saint Germain en Laye, France) and the pCMV3-CYP2D6 (Sino Biological Inc., Eschborn, Germany) plasmids were bacterially amplified and purified using Nucleobond ^®^ Xtra Midi kit (Macherey-Nagel, Hoerdt, France) and quantified by measuring the optical density at 260 nm. Transfections of human hepatoma cells with the pmax-GFP expression vector were performed using the commercially available transfection reagents Lipofectamine 3000^®^ (ThermoFisher Scientic, Waltham, MA, USA) and the liposomes prepared from the cationic lipids Syn 1 to 5 (Table 1), which belong to phosphoramidate compound family, available through the technological platform SynNanoVect (https://www.univ-brest.fr/synnanovect, accessed on 15 November 2022) from the Biogenouest network (https://www.biogenouest.org, accessed on 30 November 2022).

The liposomal solutions were prepared by the method of the lipid film hydratation. Briefly, for the preparation of liposomal solutions of Syn 1 to 5, a volume corresponding to 1.5 µmol of lipids was withdrawn from a stock solution in CHCl_3_ and placed in glass tube. The solvent was removed in vacuum in a rotavapor for at least 2 h (h) to remove all the solvent. Then, 1 mL of water was added and the solution was left at 4 °C for 24 h. Then, the solution was vortexed 3 times for 15 s (sec) and sonicated 3 times for 30 min (min) (ultrasound bath, VWR model) at 50 °C. The final concentration of the liposomal solution was 1.5 mM. 

For Syn liposomes, different amounts of transfection reagents with a constant amount of DNA were used in order to test 5 charge ratios. The charge ratio is given by R = (mass of reagent/molecular weight of reagent)/(mass of DNA/molecular weight of DNA). We used R = 1 (2 μL lipid/2 μg DNA), R = 2 (4 μL/2 μg DNA), R = 3 (6 μL/2 μg DNA) and R = 4 (12 μL/2 μg DNA).

For Lipofectamine 3000^®^, the molecular mass of the compound and/or concentration of reagent are not provided by suppliers. However, five volumes of transfection reagents (μL) were used, as recommended by the manufacturers, with a constant amount of plasmid (2 μg), giving 5 charge ratios: R = 1 (2 μL/2 μg DNA), R = 2 (4 μL/2 μg DNA), R = 3 (6 μL/2 μg DNA), R = 4 (8 μL/2 μg DNA) and R = 5 (10 μL/2 μg DNA).

The preparation of lipoplexes was similar for all reagents. For 1 well in 24-well plates, the transfection reagents and 2 μg of plasmid DNA were separately diluted in 100 μL of Opti-MEM medium (ThermoFisher Scientic, Waltham, MA, USA). The diluted transfection reagents and plasmid solutions were mixed, vortexed for 20 s and incubated at room temperature for 30 min to form lipolexes. The culture medium from the 24 well plates was discarded and renewed with 400 μL of fresh culture medium without antibiotics. The 100 μL of liposome-plasmid mixed solutions were added dropwise into the 24 well plates, which were incubated overnight with cells. The next day, the transfection medium was discarded and fresh medium containing antibiotics was added. The expression of the GFP expression was analyzed at different times post-transfection.

### 2.3. Fluorescence Microscopy and Flow Cytometry

Fluorescence microscopy was used to visualize GFP positive (GFP+) cells (Appendix A). To analyze transfection efficiencies, numbers of GFP+ cells and means of fluorescence were measured by flow cytometry. Cells were detached with trypsin, resuspended in William’s E medium containing FCS. The fluorescence intensity of 10,000 cells was analyzed with a Becton Dickinson LSRFortessa™ X-20 and the FACSDiva acquisition software (cytometry core facility of the Biology and Health Federative research structure Biosit, Rennes, France). The mean of fluorescence intensity (MFI) was measured on the GFP+ cell population (Appendix A), which reflects the relative GFP expression levels in cells. Flow cytometry data were analyzed using FlowLogic software (7.2.1 version, Inivai Technologies, Mentone Victoria, Australia).

### 2.4. Cell Viability

The effect of lipofection reagents on cell viability was established with the total cell number in each condition and expressed in percentage of the cell counts in non-transfected control cells. The cell counts were established from the cell suspensions prepared for the evaluation of the percentages of GFP+ cells by flow cytometry. The cell number was estimated using the time passage counting setting of the flow cytometer (Beckon Dickinson LSR Fortessa X-20). For this purpose, we used the singlet max time and the singlet total number values. In order to compare all the conditions, the flow rate of the cytometer was always set to 60 µL/min and the cells were always cultured and differentiated in 24 well plates, detached with the same volume of trypsin and resuspended with the same volume of culture media containing 10% FCS.

### 2.5. RNA Expression and RT-qPCR

After discarding culture medium and washing with PBS, cells were lysed and total RNA were extracted using Macherey-Nagel NucleoSpin^®^ RNA kit. Following elution, quantification of total RNAs was performed by measuring optical density at 260 nm. Then, reverse transcription was performed using 1 µg of total RNAs using the high-capacity cDNA reverse transcription kit (Applied Biosystems). RNA expression levels of specific genes (Table 2) were measured with SYBR Green PCR Master Mix kit (Applied Biosystems) using the QuantStudio^TM^ 7 flex instrument (Applied Biosystems). TATAbox Binding Protein (TBP) genes was used as reference gene.

### 2.6. Determination of Catalytic Activities of Phase I and II Enzymes by LC/MS-MS

HepaRG™ cells were incubated with a mix of phase I probe substrates, i.e., phenacetin (CYP1A2, 200 µM), bupropion (CYP2B6, 100 µM), both purchased from Sigma-Aldrich, and midazolam (CYP3A4/5, 100 µM, Pharmacopée Européenne) for 1 h, and paracetamol for phase II sulfation and glucuronidation activities for 5 h. The reaction was stopped by addition of ice-cold acetonitrile and supernatants were collected after centrifugation. The samples were stored at −20 °C until analysis.

CYP2D6 activities were evaluated by CL_int_ method using dextromethorphan as substrate and dextrorphan as metabolites (both purchased from Sigma-Aldrich, St. Louis, MO, USA). Biotransformation of dextromethorphan into dextrorphan is mediated by CYP2D6 (Km = 3.7 µM) and CYP3A4 (Km = 157 µM) [37]. CYP2D6 was identified as the dominant enzyme mediating dextrorphan formation at substrate concentrations below 10 µM to avoid CYP3A4 interaction [38]. HepaRG^TM^ were incubated with dextromethorphan (1 µM), phenacetin (CYP1A2, 1 µM), bupropion (CYP2B6, 1 µM), all, and midazolam (CYP3A4/5, 1 µM). Supernatants were collected and the reactions were stopped by addition of ice-cold acetonitrile after 5, 10, 15, 20, 30, 45 and 60 min. The samples were stored at −20 °C until analysis. The day of analysis, 10 µL aliquot of supernatant was transferred to a 96-well plate with round bottom containing 90 µL of an internal standard working solution, i.e., acetaminophen-D4, OH-bupropion-D6, dextrorphan-D3 and OH-midazolam-13C (TRC, Toronto, ON, Canada). The formed metabolites, i.e., acetaminophen, dextrorphan, 4-OH-bupropion (TRC, Toronto, ON, Canada), 1-OH-midazolam (Bertin pharma) were quantified by liquid chromatography coupled to tandem Mass Spectrometry (LC/MS-MS).

The analysis was performed using an Exion LC AC system (Applied Biosystems/Sciex, USA). Compounds were separated on a Xselect HSS T3 column (2.1 × 75 mm, 2.5 µm, Waters, Milford, MA, USA) maintained at 35 °C. The mobile phase was composed of 2 mM ammonium formate containing 0.1% formic acid (A) and acetonitrile (B). Detection was operated on a triple quadrupole mass spectrometer, AB Sciex Triple Quad 5500+ (Applied Biosystems/Sciex, Framingham, MA, USA), equipped with an electrospray ion source in positive mode. Acquisition was performed using the multiple reaction monitoring (MRM). Data processing was performed using Analyst 1.7.2 software.

Vmax was calculated using this equation:(1)$$\mathrm{Vmax}\;\left(\mathrm{pmol}.\min^{-1}.{\mathrm{mg}}^{-1}\right)=\left[\mathrm{Metabolite}\right]\;\left(µ\mathrm{mol}.{µL}^{-1}\right)\times \frac{\mathrm{volume}\;\mathrm{of}\;\mathrm{incubation}\;\left(µL\right)}{t\;(\min)\times \;\mathrm{amount}\;\mathrm{of}\;\mathrm{protein}\;\left(\mathrm{mg}\;\right)}$$

Clint was calculated using this equation:(2)$$\mathrm{Clint}\;\left(µL.\min^{-1}.{\mathrm{mg}}^{-1}\right)=\frac{\mathrm{volume}\;\mathrm{of}\;\mathrm{incubation}\;\left(µL\right)}{\ln\;\{\left[\mathrm{substrate}\right]\times \;t\;\left(\min\;\right)\}\times \;\mathrm{amount}\;\mathrm{of}\;\mathrm{protein}\;\left(\mathrm{mg}\;\right)}$$

### 2.7. DNA Replication Assay

Cells were incubated with 100 μM of 5-bromo-2-Deoxyuridine (BrdU) for 6 h. The cells were then washed 3 times with PBS and fixed for 20 min with 4.0 % paraformaldehyde diluted in PBS. The cells were then washed, permeabilized with 0.1% Triton X-100 diluted in PBS during 10 min, and treated with 0.2 M of HCl for 20 min at room temperature. Cells monolayers were incubated with anti-BrdU antibody (Abcam ab8152) diluted 1/200 in PBS with 2% BSA and 0.05% Tween20 for 2 h and then incubated with secondary antibody (DyLight 488) diluted 1/500 in PBS with 2% BSA and 0.05% Tween20 for 2 h. The cells were then washed several times with PBS. Images were acquired using a fluorescent microscope with a 20× objective. Images were further analyzed using Image J software (National Institutes of Health, Bethesda, MD, USA) to count the number of BrdU-positive cells. The percentage of proliferating cells was calculated for each condition relative to the total number of Hoechst-positive cells.

### 2.8. Protein Expression by Immunoblotting

For immunoblotting, culture medium was discarded and cells were washed with PBS and lyzed using lysis buffer: Tris 25 mM pH 7.4, 0.1% SDS, NaCl 150 mM, 1%NP-40 and 0.5% sodium deoxycholate supplemented with protease inhibitors (EDTAfree, Roche, Basel, Switzerland). Samples were then sonicated and their total protein contents were quantified using Biorad protein reagent assay. For immunoblotting, 30 µg of protein was mixed with Laemmli buffer and boiled during 10 min. The samples were next separated on NuPAGE^®^ Novex^®^ Bis-Tris 4–12% gels kit (Invitrogen) and transferred to PVDF membranes (Trans-blot^®^ Turbo™ Transfer System, Biorad, Hercules, CA, USA) before performing immunoblotting using the following primary antibodies against: GFP (anti GFP goat polyclonal antibody, sc-5384, Santa Cruz Biotechnology), CYP2D6 (anti-CYP2D6, 738667S, Cell Signaling), CYP3A4 (anti-CYP3A4, AB1254, Millipore) and HSC70 (anti-HSC, mouse monoclonal antibody, sc-7298, Santa Cruz Biotechnology, Dallas, TX, USA). Primary antibodies were detected using secondary rabbit or mouse antibodies coupled to horseradish peroxidase (HRP) (Dako, Denmark). Detection of the immune complex was performed using a chemiluminescent HRP substrate (Pierce™ ECL Substrate, Waltham, MA, USA) using the Fusion FX system (Vilber-Lourmat, Eberhardzell, Germany).

### 2.9. Statistical Analyses

Results were expressed as mean ± SEM (the standard error of the mean) of at least three independent experiments (*n*). Statistically significant variations after treatment were compared with controls using Student’s test with Excel software; ** p* value < 0.05, ** *p* < 0.01 and *** *p* < 0.001.

## 3. Results

### 3.1. Screening of Efficient and Non-Toxic Liposomes for Lipofection of Human Hepatoma Cells

In order to optimize the lipofection of hepatoma cells, we selected 5 cationic lipids, further referred as to Syn 1 to 5 in this study (see Table 1 for structures, other names and references). Syn 1 is a non-conventional lipophosphoramidate possessing two different lipid chains (non-symmetric cationic lipids), which exhibited high transfection efficiencies in different mammalian cell lines with low cytotoxicity [33]. Syn 2 is our benchmark compound, an arsenolipophosphoramidate previously produced and evaluated by our consortium (SynNanoVect platform) using different cells lines [34] including the progenitor HepaRG™ cells [26]. Syn 3 is a lipophosphoramidate-based amphiphilic compound incorporating a hydroxyl functional group in its cationic head [35]. Syn 4 is the *O*,*O*-dioleyl-*N*-(3-*N*-(*N*-methylimidazolium iodide)propylene) phosphoramidate [36]. Syn 5 is also a lipophosphoramidate with a trimethylammonium polar head [35]. Syn 2 to 5 exhibit similar lipid chains. These phosphoramidate-based lipids, previously evaluated for gene delivery, gave high transfection efficiencies using various human epithelial cells [33,34,35,36], which led us to study their ability to transfect human hepatoma.

Liposomes prepared from Syn 1 to 5 cationic lipids were first used to transiently transfect the pmax-GFP plasmid into Huh7, HepG2 (Figure 1), HBG (BC2 clone) and progenitor HepaRG™ (Figure 2) cells in order to identify the most efficient and less toxic cationic liposomes allowing GFP expression in each cell line. Transfection efficiencies and cytotoxicities were compared to those obtained with Lipofectamine^®^ 3000 reagent. Five charge ratios were used (R = 1 to 5) to study the effects of various amounts of liposomes at constant amounts of plasmid DNA on transfection efficiency and toxicity. The transfection efficiency was evaluated by fluorescence microscopy and flow cytometry (Appendix A) to quantify the percentage of GFP-positive cells and fluorescence intensities of GFP-positive cells (mean of fluorescence: MFI) reflecting the overall GFP expression and more indirectly the intracellular amounts of pmax-GFP plasmids. The highest transfection efficiencies were obtained using the Syn 1 and Syn 3 liposomes for Huh7 cells and using the Syn 2 liposome for HepG2 cells (Figure 1). The percentages of GFP+ cells reached ~50% for Huh7 at R = 2 cells and increased up to 76% for HepG2 cells at R = 1. The fluorescence intensities (MFI) were nearly equivalent between the liposomes for both Huh7 and HepG2 cells although Syn 2 liposome allowed the highest fluorescence levels in the HepG2 cells at R = 3. As previously reported for other cationic lipids [26,29], we observed that increasing charge ratios correlated with decrease in cell viabilities.

In Huh7 cells, the cell viability progressively decreased with higher charge ratios excepted with Syn 2. The HepG2 cells were very sensitive to high charge ratios with strong toxicity levels; however, at low charge ratios, the cytotoxicities remained moderate. Regardless of the charge ratio and the cell line, transfection efficiencies with Lipofectamine^®^ 3000 reagent were very high, up to 90% of GFP+ cells in both cell lines, but the cytotoxicities were also significantly increased with ~30% of viable cells at low charge ratios (Figure 1).

With phosphoramidates, the highest transfection efficiencies were obtained using the Syn 3 and Syn 1 liposome for BC2 and progenitor HepaRG™ cells, respectively (Figure 2). The percentages of GFP+ cells reached 54% for BC2 cells at R = 3 and increase up to 74% for progenitor HepaRG™ cells at R = 2 with low cytotoxicities. On the other hand, the highest fluorescence intensity was obtained using the Syn 1 reagent for progenitor HepaRG cells (R = 2) while Syn 3 and Syn 4 liposomes gave the highest fluorescence levels for BC2 cells (R = 2).

Cytotoxicity was low for BC2 cells at low charge ratios and variable cytotoxicities were observed between the liposomes when charge ratios increased (Figure 2C). Progenitor HepaRG cells were more sensitive to high charge ratios higher than R = 2 with poor viabilities below 50% compared to non-transfected cells (Figure 2F). However, toxicity remained moderate at low charge ratios as observed for Huh7 and HepG2.

While Lipofectamine^®^ 3000 reagent gave similar efficiencies of GFP expression in BC2 cells than in Huh7 and HepG2 with nearly 90% of GFP+ cells, the gene delivery in HepaRG™ cells was much lower (~30%) regardless of the charge ratio used. In addition, this reagent also triggered a strong cytotoxicity with less than 20% of viable cells in our experimental conditions of transfection.

Together, these data allowed us to identify lipophosphoramidate-based lipids and charge ratios that optimize gene delivery in hepatoma cell lines used worldwide, Huh7, HepG2, BC2 and HepaRG™ cells. For Huh7 and BC2, Syn 3 provided the best compromise between GFP expression and viability. For progenitor HepaRG™ and HepG2 cells, the highest gene deliveries were obtained with Syn 1 and Syn 2, respectively.

### 3.2. Proliferation of Differentiated HepaRG™ Hepatoma Cells

In the first part of this study, we identified Syn 1 lipid as a potent transfection agent for progenitor HepaRG™ cells. These hepatoma cells exhibit the remarkable ability to differentiate into hepatocyte-like cells in appropriate culture conditions [13,14]. The typical procedure to achieve the proper hepatocytic differentiation spans over one month of culture including the following steps (Figure 3A): (1) the active proliferation of progenitor HepaRG™ cells seeded at low density during the first week (2) a progressive quiescence taking place in the second week correlating with the commitment to the hepatocyte lineage, (3) at high density, differentiation of committed hepatocytes is further enhanced by maintaining the cells for 2 more weeks in culture medium supplemented with 2% of dimethyl sulfoxide. Despite the high differentiation of hepatocyte-like HepaRG™ cells covering nearly 90% of the hepatocyte specific functions [39], some liver genes are expressed at low levels in this cell line, which reduces their use. In order to transiently express additional genes in hepatocyte-like HepaRG™ cells, we have developed methods to transiently transfect the differentiated cells. Efficient electroporation procedures have been successfully adapted to differentiated HepaRG™ cells [26,27]. This method, however, requires to detach the cells from tissue culture dishes leading to a transient decrease in the differentiation status of the cells.

Considering the efficient gene delivery obtained in progenitor HepaRG™ cells with the Syn 1 lipid, we evaluated whether this lipophosphoramidate could also be used to transfect differentiated HepaRG™ cells without detaching cells. Transfection of the pmax-GFP plasmid into differentiated HepaRG™ cells using the Syn 1 liposome at R = 2 charge ratio produced a very low gene delivery with only 5% of GFP+ cells and a weak fluorescence intensity level both in presence of DMSO and slightly higher efficiencies up to ~10% when lipofections were performed immediately after DMSO removal (Figure 3A). These data demonstrated that Syn 1 lipid could not promote on its own an efficient gene delivery in differentiated cells. We hypothesized that the quiescence of these cells might be the main reason of this lack of efficacy since lipofection is a proliferation sensitive method of gene transfer [30,31,32].

There is a large body of evidence demonstrating that DMSO promotes the expression of liver specific functions in primary hepatocytes [40] and in HepaRG™ cells [12,15,17,41]. Interestingly, we have recently shown that DMSO inhibited proliferation of differentiated HepaRG™ cells and that removal of DMSO from the culture medium triggered a moderate but regular proliferation of differentiated HepaRG™ cells correlating with a partial decrease in the expression levels of liver specific functions [41]. These data demonstrated that differentiated HepaRG™ keep their potential of proliferation even when maintained confluent without detachment of the culture dish leading to a significant cell expansion over several days after DMSO removal [41]. We thought that the potential of differentiated HepaRG™ cells to proliferate could be used to improve the gene delivery.

It is well-established that proliferation of hepatocytes during liver regeneration is controlled by pro-inflammatory cytokines such as Interleukins 1 and 6, Tumor Necrosis Factor alpha and growth factors including Hepatocyte Growth Factor (HGF), Transforming Growth Factor alpha (TNFα) and Epidermal Growth Factor (EGF) [2,42,43,44,45] promoting the G0/G1 and G1/S transition, respectively. In a primary coculture model associating quiescent normal hepatocytes and rat liver epithelial cells, which shares phenotypical features with the differentiated hepatocyte-and cholangiocyte-like HepaRG™ cells, proliferation of hepatocytes can also be induced by mitogenic mixtures association pro-inflammatory cytokines and growth factors [2]. In order to determine whether a strong proliferation of differentiated HepaRG™ cells can be obtained by mitogenic stimulation, we prepared 3 media containing various combinations of cytokines and growth factors and evaluated the proliferation of HepaRG™ cells. These cells were first cultured at day 1 in medium without DMSO (MIL700 + ADD710) in order to trigger the cell cycle priming [41] (Figure 3B). Then, the “primed” cells were cultured at day 2 in the mitogenic media, switched back to the medium without mitogenic cocktail at day 3 and DMSO-containing medium at day 4 (Figure 3B) to maintain the differentiation status. The 3 media induced a strong DNA replication measured by BrdU incorporation (Figure 3C) although media 2 and 3 induced a slightly higher proliferative response. For medium 3 (HepaRG™ Mitogenic medium: MIL700 + ADD710 + ADD1001), we demonstrated that the cell count was doubled at day 3 (Figure 3C). The mitogenic stimulation strongly affected the morphology of the colonies of hepatocyte-like HepaRG™ cells (Figure 3D) and the staining of nuclear DNA also demonstrated the increase in cell density. Interestingly, the Hoechst-staining of the nuclear DNA in differentiated HepaRG™ cells results in much brighter signal in hepatocyte nuclei compared to that in cholangiocytes (Figure 3D) most likely because of the chromatin organization in hepatocytes. Pictures of phase contrast and Hoechst-staining suggested the strong expansion of hepatocyte colonies over proliferation of cholangiocytes in stimulated cultures. Together, these data demonstrated that removal of DMSO combined to mitogenic stimulation using the HepaRG™ mitogenic medium induced a rapid and strong proliferation of HepaRG™ cells.

### 3.3. Phenotypical Changes of HepaRG™ Hepatoma Cells during Proliferation

In order to determine if the mitogenic stimulation modulated the differentiation of HepaRG™ cells, the expressions of liver specific functions were quantified by RT-qPCR during the 3 days of the protocol (Figure 4 and Appendix A). As expected, expression of the proliferation marker CDK1 was strongly induced by the mitogenic stimulation while expression levels of liver specific functions were differentially affected during proliferation. The relative mRNA levels of CYP1A2, 2E1, 3A4, 2B6, UGT1A9, GSTA2 and albumin were significantly decreased at day 2 during mitogenic stimulation but increased again at day 3 after removal of mitogenic stimulation. In contrast, UGT 1A1, 1A6 and 2B11 gene expressions were not modulated during the proliferation of HepaRG™ cells. In addition, the amounts of CYP1A2, CYP3A4, UGT1A1, UGT2B15 and albumin mRNAs were significantly higher at day 3 than those found at day 2, which further support the conclusion that the mitogenic stimulation favored the proliferation of hepatocyte-like cells over that of cholangiocytes resulting in hepatocyte-enriched cultures after mitogenic stimulation.

These data demonstrated that induction of proliferation in differentiated HepaRG™ cells transiently decreased the relative mRNA levels of some liver specific functions evidencing a phenotypical switch and metabolic adaptation to proliferation, which resembles to the retrodifferentiation upon stimulation by cytokines and growth factors, previously described [14].

To further characterize the modulation of metabolic activities taking place during proliferation of differentiated HepaRG™ cells and the recovery after proliferation, we next measured phase I and phase II catalytic activities using different synthetic substrates, midazolam, bupropropion, phenacetin and paracetamol, at the different time points of the protocol (Figure 5). These compounds allow to quantify midazolam 1′-hydroxylase activity mainly catalyzed by CYP3A4/5, CYP2B6-dependent bupropion hydroxylase activity, phenacetin O-deethylase activity catalyzed by CYP1A2, paracetamol glucuronidation involving several enzymes (UGT1A1, UGT1A6, UGT1A9 and UGT2B15) and sulfation also catalyzed by several sulfotransferases (SULT1A1, SULT1A3/4 and SULT1E1).

All phase I activities were significantly decreased upon mitogenic stimulation at day 2, remained low after removal of the proliferation medium at day 3 (HepaRG™ Mitogenic medium MIL700 + ADD710 + ADD1001) but increased at days 4 and 5. Both bupropion hydroxylase activity and phenacetin O-deethylase activities in cells post-stimulation returned to levels similar to those measured in cells maintained in differentiation medium containing DMSO while midazolam 1′-hydroxylase activity was still lower at day 5. Paracetamol sulfation activity catalyzed by several SULT was not significantly affected by the mitogenic stimulation compared to its levels in DMSO-control cultures while glucuronidation significantly decreased only at days 2 and 3.

Together, data of mRNA expressions and catalytic activities demonstrated that, despite a short mitogenic stimulation, induction of the cell cycle in differentiated HepaRG™ cells was concomitantly accompanied by a transient and partial decrease in the expression of a subset of liver specific functions. However, HepaRG™ cells returned to their highly differentiated status within 48 to 72 h after ending the mitogenic stimulation.

### 3.4. Lipofection of Proliferating HepaRG™ Hepatoma Cells

We next evaluated whether the proliferation of differentiated HepaRG™ cells would favor their lipofection and expression of the GFP reported gene. The lipofection of differentiated HepaRG™ cells was performed using the Syn 1 reagent, at five different charge ratios ranging from R = 0.125 to R = 2, and the pmax-GFP plasmid during their stimulation by the HepaRG™ Mitogenic medium (MIL700 + ADD710 + ADD1001) (Figure 6A). As controls, lipofections were also performed in differentiated HepaRG™ cells cultured in the DMSO-containing medium (MIL700 + ADD720) and in cells cultured in medium without DMSO (MIL700 + ADD710). The transfection efficiency was analyzed by flow cytometry at day 3 (Figure 6B).

As previously observed (Figure 3A), in HepaRG™ cells maintained in DMSO-containing medium, transfection efficiency remained low with less than 10% of GFP+ cells regardless of the charge ratio. In cells cultured for 3 days in absence of DMSO, the transfection efficiency was significantly higher than that of cells maintained in DMSO-containing medium, possibly because DMSO-removal induced a moderate cell proliferation [41]. In this culture condition, the number of GFP+ cells increased with higher charge ratio with up to 25% of transfected cells at R = 2.

When cells were stimulated, the percentages of GFP+ cells were strongly enhanced in a charge ratio-dependent manner (Figure 6B). The highest percentages were achieved with the R = 2 charge ratio allowing ~48% of GFP+ cells to express. Interestingly, the intensities of fluorescence were much higher in stimulated cells than in the control conditions suggesting that proliferation strongly increased the intracellular amounts of plasmid and/or improved the synthesis of GFP.

Time-course expression of GFP was studied by Western blot in cells maintained in DMSO-containing culture medium and in cells transfected by lipofection during proliferation induced by the mitogenic additive (Figure 6C). As expected, GFP was not detectable at day 2 of the lipofection protocol while at day 3 GFP was expressed at high levels, which was confirmed by fluorescence microscopy to visualized GFP+ cells (Figure 6D). The high expression levels of GFP at day 3 was, however, transient since protein expression progressively decreased at days 4 and 5. Together, these data demonstrated that mitogenic stimulation of HepaRG™ cells strongly enhanced the lipofection efficiencies and that enforced expression of GFP from plasmid vectors was transient for 24 to 48 h after transfection.

This optimized procedure of lipofection was used to express the cytochrome P450 2D6 (CYP2D6), which was previously reported to be detected at very low levels in differentiated HepaRG™ cells with estimated mRNA levels reaching ~0.5 to 1% of those found in human hepatocytes isolated from liver biopsies [15]. In order to confirm and extend these data, we compared CYP2D6 mRNA levels in human livers, freshly isolated and primary cultures of human hepatocytes, as well as progenitor and differentiated hepatocyte-like HepaRG™ cells (Figure 7A). In these different samples, we also analyzed the expression of CYP2E1, which is known to be expressed in differentiated HepaRG™ cells [24]. As expected, we confirmed that CYP2D6 mRNAs were barely detectable in both progenitor and differentiated HepaRG™ cells while highly expressed in human samples, which further demonstrated the very low expression of CYP2D6 in HepaRG cells. In contrast, CYP2E1 mRNA expression was induced in differentiated HepaRG™ cells.

In addition, we measured CYP2D6-dependent dextromethorphan O-demethylase catalytic activities in two large cohorts of 163 samples of human hepatocytes and 187 batches of hepatocyte-like HepaRG™ cells produced by the company Biopredic International (Figure 7B). In human hepatocytes, known to present several CYP2D6 polymorphisms, CYP2D6-dependent dextromethorphan O-demethylase catalytic activities exhibited large differences with a ~30-fold variation between the lowest and highest expression values most likely because of the genetic differences between individuals as well as the liver pathologies of the donors from which the biopsies and hepatocytes were obtained from (Figure 7B). In addition, our data also demonstrated that CYP2D6 catalytic activities were significantly much weaker in HepaRG™ cells compared to those measured in human hepatocytes, in the range of the lowest levels detected in human hepatocytes (Figure 7B).

Then, we performed lipofection of differentiated HepaRG™ cells during proliferation using Syn 1 lipid and CYP2D6 encoding plasmid (Figure 7C). As previously observed with the GFP expression (Figure 6C), CYP2D6 protein was expressed at day 3 of the stimulation protocol and its levels progressively decreased at days 4 and 5 (Figure 7C). At days 3 to 5, CYP2D6 was functional as demonstrated by the strong CYP2D6-dependent dextromethorphan O-demethylase catalytic activity measured in transfected cells (Figure 7D).

We next measured midazolam 1′-hydroxylase activity mainly catalyzed by CYP3A4/5, CYP2B6-dependent bupropion hydroxylase activity and phenacetin O-deethylase activity catalyzed by CYP1A2 in differentiated HepaRG cells and cells transfected by CYP2D6 at days 2 to 5 during and after proliferation (Figure 7D). These data showed that these CYP-dependent enzymatic activities were not significantly affected by the expression of CYP2D6 and, more important, that CYP2D6-dependent dextromethorphan activity was detected concomitantly to the other CYP activities, which demonstrated that we successfully achieved enforced expression of CYP2D6 while maintaining expression of other drug metabolism enzymes.

## 4. Discussion

Using liposomes prepared from phosphoramidate-based cationic lipids, which had previously given high transfection efficiencies with low toxicities in various human epithelial cells [33,34,35,36], our first objective was to identify the lipids that would allow the best lipofection levels of the widely used HepG2, HuH7, BC2 and HepaRG™ human hepatoma cells.

Our data demonstrated that Syn 3 provided the best GFP reporter expression and viability for both Huh7 and BC2 cells while Syn 1 and Syn 2 allowed the highest gene deliveries for progenitor HepaRG™ and HepG2 cells, respectively. Interestingly, although the in vitro cell models used in these experiments were all human hepatoma cell lines, we could not identify a unique cationic lipid suitable for gene delivery in the four cell lines. These data further emphasized the differences in gene delivery efficiencies observed between cell types, even from the same cell lineage. For instance, in a detailed study, Aliabadi and co-authors have shown the similarities but also differences in siRNA delivery into multiple breast cancer cell lines using a series of lipopolymers [46], probably related to the structure of the lipopolymers used for siRNA complexation, their interaction with the cells and the endocytosis activities of the gene delivery systems by the different cell lines. Similarly, we also recently showed that the overall endocytosis activity considerably varies between hepatoma cells [47], which may partially explain the differences in the transfection efficiencies observed between hepatoma cell lines with the same cationic liposomes.

Large progress in understanding the mechanisms of nanovector’s cell uptake in the recent years have clearly evidenced the biological barriers strongly affecting the non-viral gene delivery. These different limiting steps have been described in the past decades following the synthesis of a plethora of cationic lipids and polymers and their use for the delivery of nucleic acids both in vitro and in vivo in multiple proliferating and quiescent cell types [48,49,50]. These physical and biological barriers to efficient non-viral gene delivery include: (1) the physicochemical features and colloidal stability of nucleic acids (DNA or RNA) bound to their delivery nanoparticles, in vitro in culture media and in vivo in “biological fluids” prior to cell uptake, (2) cellular internalization by endocytosis or phagocytosis, (3) endosomal escape, cytosolic trafficking and/or nuclear translocation, and 4) mRNA transfer to the endoplasmic reticulum for efficient translation or nuclear localization of plasmids for recruitment by the transcription machinery.

Numerous cationic lipids and polymers have been optimized to improve the colloidal stability, cell internalization of plasmid/cationic vector complexes and transfection effectiveness while limiting the cytotoxicity [51]. Using these lipids and polymers, the cell uptake of nucleic acid-cationic complexes across the plasma membrane does not longer appear as the main limiting step in gene delivery, since high amounts of complexes are detected intracellularly [52,53]. Interestingly, this internalization process mainly occurring via clathrin-mediated, caveola-mediated endocytosis, or macro and micropinocytosis [50] is highly modulated through the opsonization of complexes by plasma proteins and the formation of a protein corona at the surface of nanovectors [54,55]. This phenomenon observed with all drug delivery systems in vivo but also in vitro when cells are cultured in presence of bovine and human serum, affects cell uptake [56] and could also impact the intracellular behavior of the complexes.

A first major bottleneck in gene delivery is the endosomal escape of nucleic acids and their intracellular traffic. Following endocytosis, gene delivery systems proceed through an endosomal trafficking process with only a small fraction of nucleic acids that escape the degradation in lysosomes to become active biologically for protein expression (plasmid DNA and mRNA) or gene repression (siRNA) [48,49,50]. From the small fraction of nucleic acid that is able to escape this endosome-lysosome pathway, the cytoplasm and nuclear envelope constitute other major physiological barriers at least for gene delivery based on the use of plasmid DNA. Indeed, plasmids translocate to the nucleus passively during mitosis in proliferating cells [57,58] and in a much weaker extend through the nuclear pore complex in poorly dividing cells [59]. Thus, lipofection remains a delivery technology for protein encoding DNA templates much more efficient in proliferating than in quiescent cells [31,32,60,61]. However, the emergence of stabilized mRNA, produced by in vitro transcription using modified base pairs, provided a novel paradigm for lipid-mediated mRNA delivery more efficient than lipofection of DNA in non-dividing cells [62], since mRNA are readily available for translation in cytosol and do not require nuclear translocation. This highly efficient lipid-based delivery of mRNA [63] along with the large-scale production of mRNA lipid nanoparticles by microfluidics [64] offered a therapeutic answer to the covid-19 pandemic by providing the first mRNA vaccine in humans used worldwide [65].

In our study, we confirmed that the proliferation status was crucial in the transfection of DNA in HepaRG™ cells mediated by phosphoramidate-based lipids. While Syn lipids allowed high gene delivery in proliferating hepatoma cells including progenitor HepaRG™ cells, in contrast, the GFP reporter expression was very low in quiescent differentiated HepaRG™ cells. It is well admitted that quiescent hepatocytes in primary culture are difficult to transfect [66,67]. However, murine hepatocytes upon mitogenic stimulation undergo 1 or 2 rounds of cell cycle, which allowed significant transfection levels reaching percentages up to ~30% [28]. In contrast, the lipid- and polymer-mediated gene transfer in human hepatocytes that poorly proliferate, even after mitogenic stimulation, remains extremely low as observed for quiescent differentiated HepaRG™ cells [27]. In a previous report [29], we established a protocol of lipofection for differentiated HepaRG™ cells using liposomes prepared from diether-NH_2_ combined with Egg Phosphatidyl Choline (EPC). We also took advantage of the procedure developed in our laboratory to selectively detach differentiated hepatocyte-like HepaRG™ cells from cholangiocytes and to plate enriched cultures of hepatocyte-like cells [13,27], which show a transient and moderate proliferation activity during 48 h [13]. In this setting, we found that ~30% of hepatocyte-like HepaRG cells expressed the GFP reporter protein following lipofection using EPC/diether liposomes [29]. Nevertheless, this procedure is difficult to handle with time-consuming culture manipulations and limited transfection efficiencies.

Herein, we reported a more straightforward procedure to achieve the robust proliferation of differentiated HepaRG™ cells without cell detachment based on the design of a mitogenic culture medium, which stimulates re-entry in the cell cycle, DNA replication and cell duplication. The mitogenic stimulation was characterized by induction of the cell cycle marker CDK1 and changes in the expression levels and catalytic activities of some liver specific functions. Among those hepatic genes, phase I enzymes such as CYP1A2, 2E1 and 3A4 are partially down-regulated by the mitogenic stimulation in agreement with a previous report showing that pro-inflammatory cytokines such as Interleukin 6 and Tumor Necrosis Factor alpha triggered the retrodifferentiation of hepatocyte-like HepaRG™ cells [14]. Interestingly, the Hoechst-staining of the nuclear DNA and expression levels of liver-specific functions in HepaRG™ cells strongly suggested that the mitogenic stimulation favored the proliferation of hepatocyte-like cells over that of cholangiocytes. Indeed, the “basal” medium (MIL700) used for HepaRG™ cells contains high amounts of insulin and hydrocortisone hemisuccinate, two soluble factors favoring the differentiation in the hepatocytic lineage. In addition, we used different mixtures of pro-inflammatory cytokines and growth factors known to be major regulators of the hepatocyte proliferation both in vitro and in vivo during liver regeneration [2,42,43,44,45]. The composition of culture media used in our study may thus favor proliferation of hepatocyte-like cells resulting in hepatocyte-enriched cultures at days 4 and 5 after mitogenic stimulation. Independently, it was reported that some cytokines down-regulate expression of major CYP in quiescent human and rodent hepatocytes both in vitro and in vivo in absence of proliferation [68,69,70], which demonstrated that these soluble mediators regulate a subset of drug-metabolism enzymes regardless of the induction of DNA replication and mitosis.

Our data also demonstrated that cytokines do not induce a complete dedifferentiation program in HepaRG™ cells during a short 24 h-mitogenic stimulation since expression levels of UGT 1A1, 1A6 and 2B11 mRNA were not significantly modified and paracetamol sulfation activities remained unchanged during proliferation. Moreover, removal of the mitogenic stimulation was followed by the rapid recovery of the liver specific functions that were down-regulated during proliferation. Together, these data demonstrate that differentiated HepaRG™ cells provide a suitable model to study the equilibrium between differentiation and proliferation in hepatocytes, which is tightly regulated by cytokines, growth factors, cell–cell and cell–extracellular matrix interactions [2,71]. Furthermore, the model of proliferating differentiated HepaRG™ cells will be of interest to identify chemicals that require bioactivation to cause genotoxicity in metabolically competent cells exposed prior to proliferation and high-throughput analysis of the mutational events integrated into the nuclear DNA during replication [72,73].

The active proliferation of differentiated HepaRG™ cells for a short period of time enabled much higher lipofection efficiencies using Syn 1 lipophosphoramidate lipid compared to transfection levels in non-stimulated cells, which further confirmed the cell cycle dependence of DNA transfer in this cell type. With charge ratios between 1 and 2, Syn 1 liposomes allowed 30 to 50% of GFP+ cells with maximum expression levels at day 3 of the lipofection procedure. We took advantage of this efficient lipofection protocol to achieve the expression of the CYP2D6, which is expressed at very low levels in HepaRG™ cells [15]. As observed with the GFP, CYP2D6 was well expressed at day 3 and its level and catalytic activity, measured with the dextromethorphan, progressively decreased within 3 days after lipofection. Despite a transient expression, efficient CYP2D6 lipofection was successfully obtained, which opens new means to evaluate CYP2D6-dependent drug metabolism in transfected HepaRG™ cells.

## 5. Patents

The HepaRG™ cell line is patented (PCT/FR02/02391 July 8, 2002), and licensed to Biopredic^Int^, which produces and commercializes the cells and associated culture media. The academic investigators (Anne Corlu and Pascal Loyer) of the Institute NUMECAN (Nutrition Metabolisms and Cancer, Inserm UMR-S 1241, INRAE UMR-A 1341, Univ Rennes, F-35000 Rennes, France) identified a mitogenic mixture to stimulate the proliferation of differentiated hepatocyte-like HepaRG™ cells, protected by a know-how which define the combination and concentrations of cytokines and growth factors. The know-how was licensed to Biopredic^Int^ to commercialize this new product under the following reference: Differentiated HepaRG™ Mitogenic medium supplement (ADD1001).

## Figures and Tables

**Figure 1 cells-11-03904-f001:**
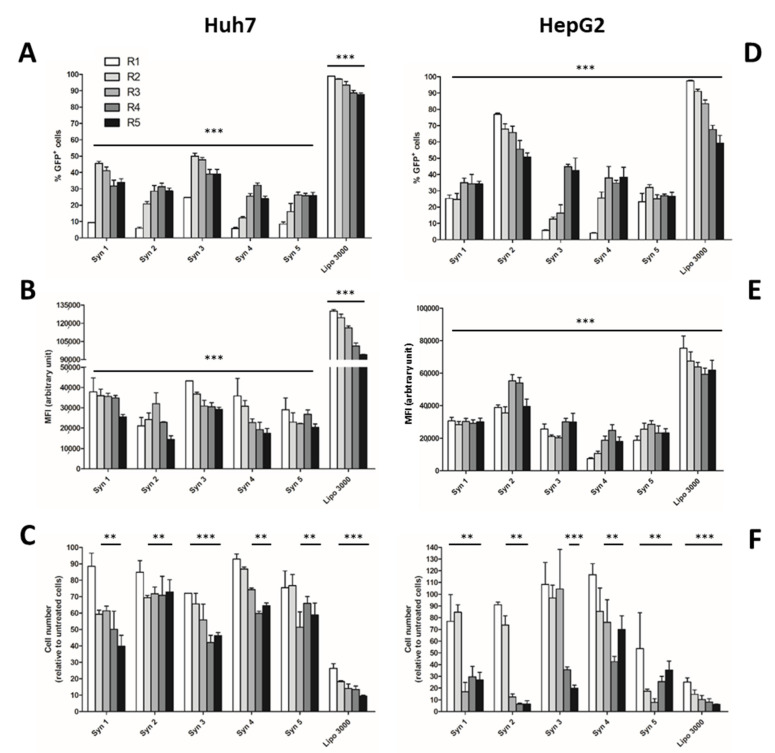
Transient transfection of Huh7 and HepG2 cells. The cationic liposomes Syn 1 to Syn 5 and Lipofectamine^®^ 3000 were used to transfect the pmax-GFP plasmid into Huh7 and HepG2 cells at five different charge ratios (*R*). The transfection efficiency was quantified by flow cytometry to determine the percentage of GFP+ cells (**A**,**D**) and the fluorescence intensity (**B**,**E**). The cell viability was established with the total cell number in each condition and expressed in percentage of cell number compared to non-transfected control cells (**C**,**F**). Since non-transfected control cells did not express GFP, all lipofection reagents allowed detection of GFP+ cells (≥5%) and intensities of fluorescence (MFI) significantly higher than those of non-transfected cells (*** *p* > 0.001). For the cell viability assay, statistically significant differences are presented as ** *p* < 0.01, *** *p* < 0.001, for non-transfected versus transfected conditions.

**Figure 2 cells-11-03904-f002:**
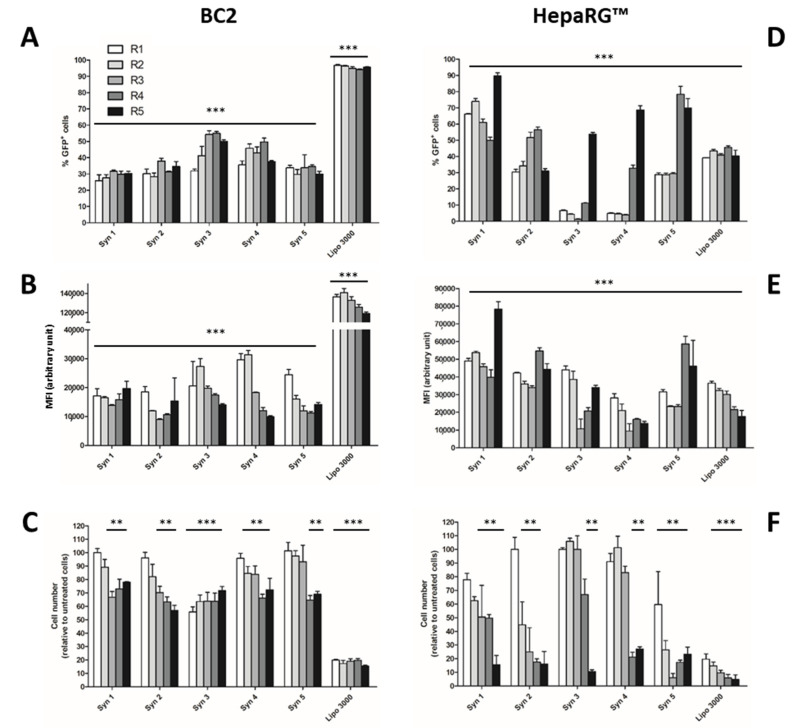
Transient transfection of BC2 and progenitor HepaRG™ cells. The cationic liposomes Syn 1 to Syn 5 and Lipofectamine^®^ 3000 were used to transfect the pmax-GFP plasmid into BC2 and progenitor HepaRG™ cells at five different charge ratios (*R*). The transfection efficiency was quantified by flow cytometry to determine the percentage of GFP+ cells (**A**,**D**) and the fluorescence intensity (**B**,**E**). Since non-transfected control cells did not express GFP, all lipofection reagents allowed detection of GFP+ cells (≥5%) and intensities of fluorescence (MFI) significantly higher than those of non-transfected cells (*** *p* > 0.001). The cell viability was estimated by counting the total cell number in each condition using the flow cytometer. The results are expressed in percentage of cell number compared to non-transfected control cells (**C**,**F**). For the cell viability assay, statistically significant differences are presented as ** *p* < 0.01, *** *p* < 0.001, for non-transfected versus transfected conditions.

**Figure 3 cells-11-03904-f003:**
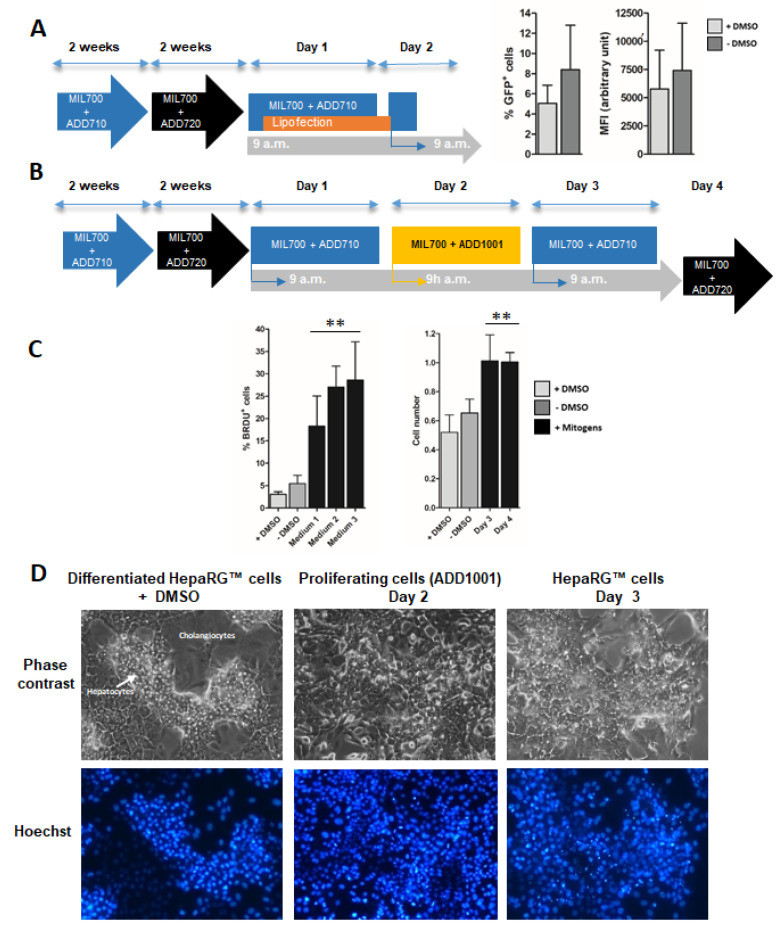
Transfection and induction of proliferation in differentiated HepaRG™ cells. The protocol of differentiation is depicted by the scheme (A): Progenitor HepaRG™ cells were differentiated over a month of culture using the culture medium (MIL700) successively supplemented with additives ADD710 and 720. Syn 1 liposome was used to transfect the pmax-GFP plasmid in differentiated HepaRG™ cells (R = 2). The transfection efficiency (% of GFP+ cells) and fluorescence intensity (MFI) were evaluated by flow cytometry (**A**). Proliferation of differentiated HepaRG™ cells was stimulated during 24 h by culturing the cells in novel media (media 1, 2 and 3) containing different combinations of growth factors with protocol depicted in scheme (**B**). Mitogenic stimulation was achieved by culturing the differentiated HepaRG™ cells in culture medium MIL700 + ADD710 for 1 day, then in the same medium supplemented by the mitogenic mix (ADD1001). At day 3, cells were maintained in medium MIL700 + ADD710 without growth factors and switched back to the differentiation medium MIL700 + ADD720. Cell proliferation was measured by BrdU incorporation (6-h incubation at day 2 during mitogenic stimulation) and overall cell counts were determined at days 3 and 4 in cultures incubated with medium 3 (**C**). Phase contrast and Hoechst staining of nuclear DNA showed the impact of medium 3 on the morphology of HepaRG™ cell monolayer and the density of nuclei in hepatocytes colonies, respectively (**D**). Statistics: Significantly different for non-stimulated versus stimulated conditions with ** *p* < 0.01.

**Figure 4 cells-11-03904-f004:**
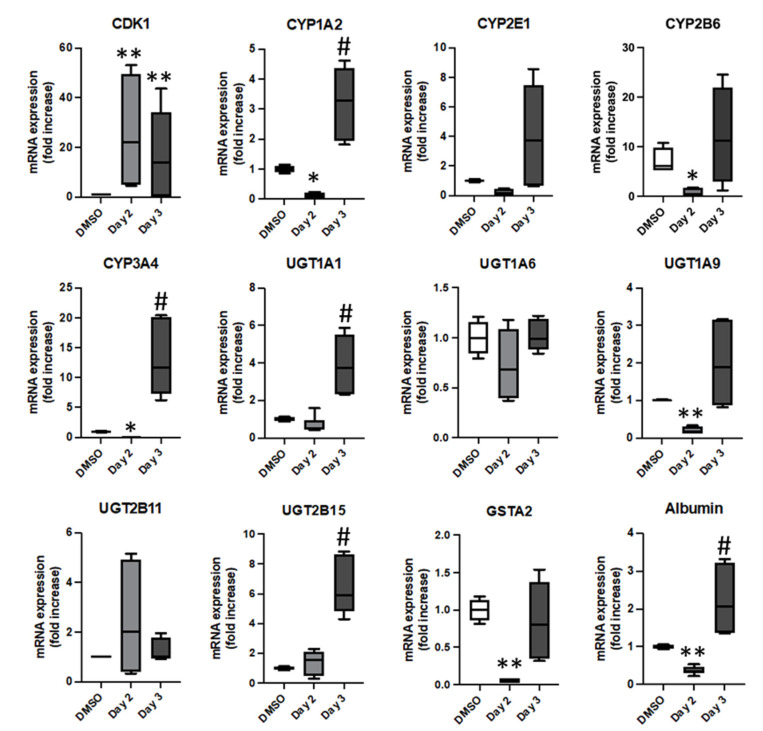
Expression of the proliferation marker CDK1 and liver specific functions during proliferation of differentiated HepaRG™ cells. The relative mRNA expression level of CDK1 and hepatic markers (albumin, CYP1A2, CYP2E1, CYP3A4, UGT1A1, UGT1A6, UGT1A9, UGT2B6, UGT2B15, UGT2B11 and GSTA2) were measured by RT-qPCR during proliferation (protocol from Figure 3B). Analyses of gene expressions were performed in cells always kept in the differentiation medium (MIL700 + ADD720 containing DMSO (DMSO on the chats), and in stimulated cells at days 2 and 3 of the proliferation protocol. Statistics: Significantly different for non-transfected (DMSO) versus transfected conditions at day 2 as * *p* < 0.05, ** *p* < 0.05, and between non-transfected (DMSO) versus transfected conditions at day 3 # *p* < 0.05, mean of 3 to 4 independent experiments.

**Figure 5 cells-11-03904-f005:**
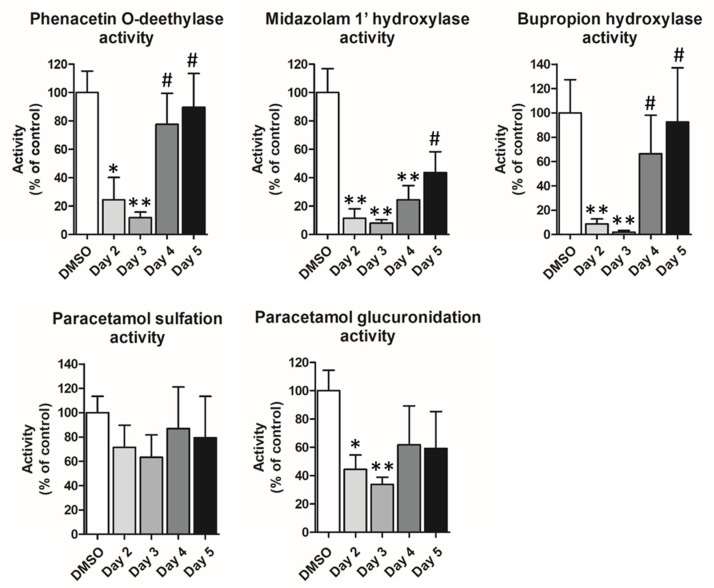
Catalytic activities of phase I and phase II enzymes during proliferation of differentiated HepaRG™ cells. The Vmax of phase I and phase II enzymes were measured by LC/MS-MS using synthetic substrates: midazolam, bupropion, phenacetin and paracetamol, in HepaRG™ cells that were not stimulated (DMSO) representing the standard of differentiation, and at different time point of the protocol described in Figure 3. Statistics: Significantly different between transfected conditions at days 2 to 4 versus non-transfected (DMSO) (* *p* < 0.05 and ** *p* < 0.01) and between transfected conditions at days 4 to 5 versus transfected conditions at days 2 to 3 (# *p* < 0.05).

**Figure 6 cells-11-03904-f006:**
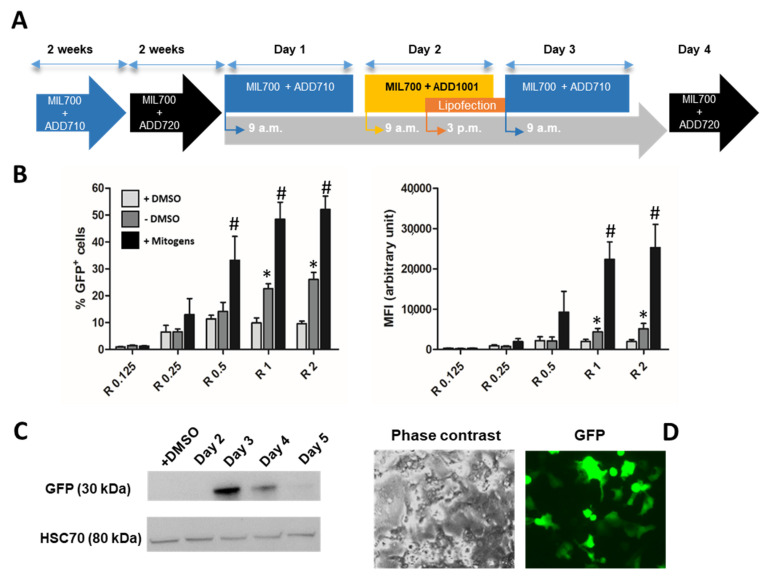
Lipofection of differentiated HepaRG™ cells during proliferation using Syn 1 lipid and GFP expression. HepaRG™ cells were cultured during 4 weeks using protocol of proliferation and differentiation described in material and methods and transfected by lipofection using Syn 1 lipid at day 2 during mitogenic stimulation (**A**). Transfection efficiency was quantified by flow cytometry to determine the percentages of GFP+ cells and the fluorescence intensities (MFI) in differentiated cells maintained in presence (+DMSO, i.e., MIL700+ADD720) or absence of DMSO (-DMSO, i.e., MIL700+ADD710) and cultures stimulated with the mitogenic additive (i.e., ADD1001) at different charge ratio ranging from R = 0.125 to R = 2 (**B**). Statistics: Percentages of GFP+ cells and MFI significantly different between cells cultured in presence and absence of DMSO (* *p* < 0.05) and between unstimulated and mitogen-stimulated cells (# *p* < 0.05). Expression of GFP was studied by Western blot (**C**). Phase contrast and fluorescence microscopy of HepaRG™ cells transfected with GFP encoding plasmid (**D**).

**Figure 7 cells-11-03904-f007:**
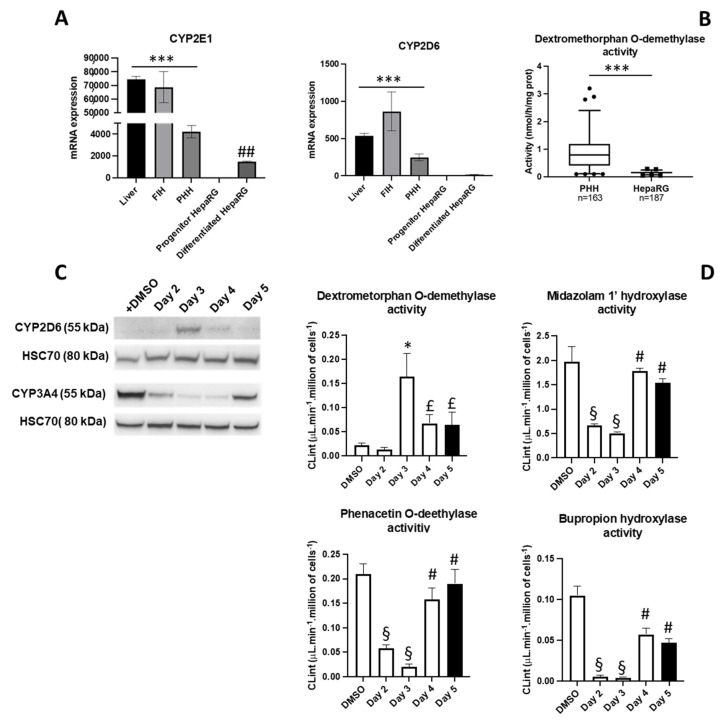
Expression and catalytic activities of CYP2D6 in human hepatocytes and differentiated HepaRG™ cells. The relative amounts of CYP2E1 and CYP2D6 mRNA were investigated by RT-qPCR in biopsies of human normal livers (*n =* 5), freshly isolated human hepatocytes (FIH, *n =* 3), primary human hepatocytes cultured for 48h (PHH, *n =* 4), progenitor (*n =* 3) and differentiated HepaRG™ cells (*n =* 3) (**A**). Significantly different between human livers and primary hepatocytes *versus* HepaRG™ cells (* *p* < 0.05) and between differentiated hepatocyte-like and progenitor HepaRG™ cells (# *p* < 0.05). CYP2D6-dependent dextromethorphan O-demethylase catalytic activities were measured in primary human hepatocytes (*n =* 163) cells and differentiated HepaRG™ cells (*n =* 187) (**B**). Significantly different *** *p* < 0.001. Expression of CYP2D6 and CYP3A4 was studied by Western blot (**C**) and CYP2D6-dependent dextromethorphan O-demethylase, CYP3A4/5-dependant midazolam 1′-hydroxylase, CYP1A2-phenacetin O-deethylase and CYP2B6-dependent bupropion hydroxylase catalytic activities (CLint) were measured in differentiated hepatocyte-like HepaRG™ cells and transfected cells at days 2 to 5 (**D**). Dextromethorphan O-demethylase activities significantly different between transfected conditions at day 3 versus non-transfected (DMSO) and transfected cultures at day 2 (* *p* < 0.05), between transfected cells at days 4 or 5 versus transfected cultures at day 2 (£ *p* < 0.05), other catalytic activities significantly different between cultures at days 2 and 3 versus non-transfected (DMSO) cells (§ *p* < 0.05) and between cultures at days 4 and 5 versus cultures at days 2 and 3 (# *p* < 0.05).

**Table 1 cells-11-03904-t001:** Structures of the cationic lipids and characteristics of the corresponding liposomes: [a] Hydrodynamic diameter (Dh) in nanometers (nm) measured by DLS, [b] Polydispersity index (PDI) of the liposome size measured from DLS, [c] Zeta potential (ζ) in milliVolts (mV) measured using a 3000 Zetasizer (Malvern Instruments) at 25 °C.

Lipid Names	Chemical Structures	Dh ^[a]^ [nm]	PDI ^[b]^	^[c]^ [mV]	References
Syn 1	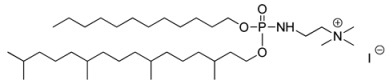	270 ± 108	0.23	+49 ± 6	[33]
Syn 2/KLN47	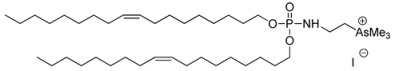	201 ± 108	0.197	+34 ± 7	[34]
Syn 3/BSV107	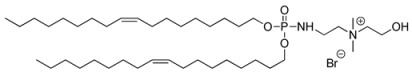	217 ± 113	0.232	+31 ± 9	[35]
Syn 4/KLN25	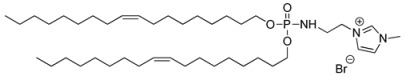	90 ± 61	0.348	+52 ± 9	[36]
Syn 5/BSV36	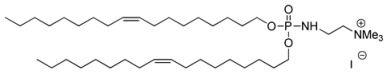	197 ± 100	0.385	+41 ± 9	[35]

**Table 2 cells-11-03904-t002:** List of forward and reverse primers used for RT-qPCR analyses.

Genes	Forward	Reverse
CDK1	CCCTTTAGCGCGGATCTACC	AGGAACCCCTTCCTCTTCACT
Albumin	TGCTTGAATGTGCTGATGACAGG	AAGGCAAGTCAGCAGGCATCTCATC
UGT1A1	TGACGCCTCGTTGTACATCAG	CCTCCCTTTGGAATGGCAC
UGT1A6	CAGTGCCGTATGACCAAGAA	GTCTGAGGAGCAGTTAGGAATG
UGT1A9	TGTCTTTAAACAAACTCCTGCAA	TGGAAAGCACAAGTACGAAGTATATA
UGT2B11	AGGTTCTGTGGAGATTTGACG	TGCCTCATAGATGCCATTGG
UGT2B15	GTGTTGGGAATATTATGACTACAGTAAC	GGGGTTAAATAGTTCAGCCAGT
GSTA2	TGCAACAATTAAGTGCTTTACCTAAGTG	TTAACTAAGTGGGTGAATAGGAGTTGTATT
CYP1A2	GGTTCAAGGCCTTCAACCAG	ACATGAGGCTCCAGGAGATG
CYP2B6	TTCCTACTGCTTCCGTCTATC	GTGCAGAATCCCACAGCTCA
CYP2E1	TTGAAGCCTCTCGTTGACCC	CGTGGTGGGATACAGCCA
CYP2D6	TAAGGGAACGACACTCATCAC	TCACCAGGAAAGCAAAGACAC
CYP3A4	CTTCATCCAATGGACTGCATAAA	TCCCAAGTATAACACTCTACACACACA
CK19	TTTGAGACGGAACAGGCTCT	AATCCACCTCCACACTGACC
GFP	ACAACAGCCACAACGTCTAT	GGGTGTTCTGCTGGTAGTG
TBP	GAGCTGTGATGTGAAGTTTCC	TCTGGGTTTGATCATTCTGTA

## Data Availability

Data supporting reported results are publicly available at Mendeley Data (https://data.mendeley.com/, accessible from 28 February 2023, doi: 10.17632/g437rkp7h6.1).

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
