# Peer review of "Liposome-Mediated Gene Transfer in Differentiated HepaRG™ Cells: Expression of Liver Specific Functions and Application to the Cytochrome P450 2D6 Expression"

_cells, 2022, doi:10.3390/cells11233904_

Round 1

Reviewer 1 Report

In the present manuscript by Vlach et al. tested different liposome-based transfection reagents for cell lines of hepatic origin showing differences depending on the cell line. They selected Syn-1 as the best transfection reagent for HepaRG. As expected, transfection efficacy was significantly when the cells are differentiated. To overcome this limitation the authors, induce cell division in differentiated cells using a mitogenic cocktail and this maneuver improved the transduction efficiency. The goal of this study is to increase the expression of proteins that are key in hepatocytes metabolic functions and that are low in HepaRG like CYP2D6. However, as shown by the author the expression of the transgenic protein in very short. Furthermore, the impact of CYP2D6 over hepatocyte functionality was not evaluate reducing the interest of the manuscript.

Furthermore what it is intriguing is the fact that after the induction of proliferation by the mitogenic agents and posterior redifferentiation there’re is a clear elevation of hepatocytes markers that should be discussed. To better understand the results, it is essential to know at least the composition of the mitogenic cocktail.

Additionally, there are several aspects that required authors attention.

Review sentence in line 117.

Since there is a lot of variability in the data, it will be useful to see the bars and the individual values in the experiments.

It would be good to include the expression levels of CYP2D6 in all the samples.

In figure 6 the graph showing CYP2D6-dependent dextromethorphan O-demethyl-ase catalytic activities should be panel D.

Reviewer 2 Report

In the current manuscript, the authors investigated the transfection efficiencies and toxicity of 5 cationic lipids and and found that Syn1-based nanovector had highest transfection efficiencies in progenitor HepaRGTM cells. However, the transfection efficiency of Syn1-bsed nanovector was very poor in the differentiated HepaRGTM cells. Take advantage of the transient proliferation of differentiated HepaRGTM cells, the authors engineered a procedure for gene delivery using Syn1-based liposomes in differentiated HepaRGTM cells. The procedure established in the current study is meaningful for drug metabolism studies. Specific comments follow below.

1. Statistics are needed for Fig 1A and 1D.

2. Representative fluorescent images of GFP in Syn1-5 and Lipo 3000 transfected cells are needed.

3. What is the mRNA expression of CYP2B6 and CYP2D6 during the protocol of proliferation described in Figure 3B?

4. In Figure 4, the expression of CYP1A2, CYP3A4 and UG1A1 at day 3 seems higher than the DMSO groups. How to explain this phenomenon? The statistics are needed.

Reviewer 3 Report

The manuscript is very thorough and offers some value to practitioners interested in using the HepaRG cell line rather than primary hepatocytes for ADME/Tox studies. By offering a means of supplementing expression of underrepresented DMEs in this cell line, outcomes should be more comparable to those with primary hepatocytes.

While figures 1 and 2 show viable cell number data, the method for counting viable cells is only vaguely explained in the legend to figure 2. Therefore, this reviewer requests adding language to the methods section that provides a detailed explanation of the cell viability assessment method.

Line 614 says "micro- and micropinocytosis". Perhaps the authors meant to say "macro- and micropinocytosis"

Round 2

Reviewer 1 Report

the authors have properly answer all reviewer's concer except for the composition of the mitogenic cocktail.